# BONGARD-LOGO: A New Benchmark for Human-Level Concept Learning and Reasoning

**Weili Nie**
Rice University
wn8@rice.edu

**Zhiding Yu**
NVIDIA
zhidingy@nvidia.com

**Lei Mao**
NVIDIA
lmao@nvidia.com

**Ankit B. Patel**
Rice University
Baylor College of Medicine
abp4@rice.edu

**Yuke Zhu**
UT Austin
NVIDIA
yukez@cs.utexas.edu

**Animashree Anandkumar**
Caltech
NVIDIA
anima@caltech.edu

## Abstract

Humans have an inherent ability to learn novel concepts from only a few samples and generalize these concepts to different situations. Even though today's machine learning models excel with a plethora of training data on standard recognition tasks, a considerable gap exists between machine-level pattern recognition and human-level concept learning. To narrow this gap, the Bongard problems (BPs) were introduced as an inspirational challenge for visual cognition in intelligent systems. Despite new advances in representation learning and learning to learn, BPs remain a daunting challenge for modern AI. Inspired by the original one hundred BPs, we propose a new benchmark BONGARD-LOGO for human-level concept learning and reasoning. We develop a program-guided generation technique to produce a large set of human-interpretable visual cognition problems in action-oriented LOGO language. Our benchmark captures three core properties of human cognition: 1) context-dependent perception, in which the same object may have disparate interpretations given different contexts; 2) analogy-making perception, in which some meaningful concepts are traded off for other meaningful concepts; and 3) perception with a few samples but infinite vocabulary. In experiments, we show that the state-of-the-art deep learning methods perform substantially worse than human subjects, implying that they fail to capture core human cognition properties. Finally, we discuss research directions towards a general architecture for visual reasoning to tackle this benchmark.

## 1 Introduction

Human visual cognition, a key feature of human intelligence, reflects the ability to learn new concepts from a few examples and use the acquired concepts in diverse ways. In recent years, deep learning approaches have achieved tremendous success on standard visual recognition benchmarks [1, 2]. In contrast to human concept learning, data-driven approaches to machine perception have to be trained on massive datasets and their abilities to reuse acquired concepts in new situations are bounded by the training data [3]. For this reason, researchers in cognitive science and artificial intelligence (AI) have attempted to bridge the chasm between human-level visual cognition and machine-based pattern recognition. The hope is to develop the next generation of computing paradigms that capture innate abilities of humans in visual concept learning and reasoning, such as few-shot learning [4, 5], compositional reasoning [6, 7], and symbolic abstraction [8, 9].

The deficiency of standard pattern recognition techniques has been pinpointed by interdisciplinary scientists in the past several decades [3, 10]. Over fifty years ago, M. M. Bongard, a Russian computer

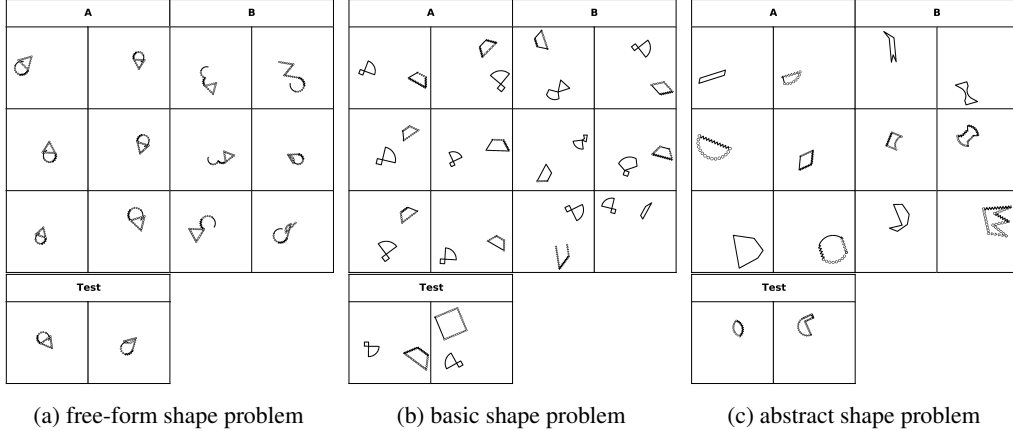

|  (a) free-form shape problem | (b) basic shape problem | (c) abstract shape problem |

Figure 1: Three types of problems in BONGARD-LOGO, where (a) the free-form shape concept is a sequence of six action strokes forming an "ice cream cone"-like shape, (b) the basic shape concept is a combination of "fan"-like shape and "trapezoid", (c) the abstract shape concept is "convex". In each problem, set $\mathcal{A}$ contains six images that satisfy the concept, and set $\mathcal{B}$ contains six images that violate the concept. We also show two test images (left: positive, right: negative) to form a binary classification task. All the underlying concepts in our benchmark are solely based on shape strokes, categories and attributes, such as convexity, triangle, symmetry, etc. We do not distinguish concepts by the shape size, orientation and position, or relative distance of two shapes.

scientist, invented a collection of one hundred human-designed visual recognition tasks, now named the Bongard Problems (BPs) [11], to demonstrate the gap between high-level human cognition and computerized pattern recognition. Several attempts have been made in tackling the BPs with classic AI tools [12, 13], but to date we have yet to see a method capable of solving a substantial portion of the problem set. BPs demand a high-level of concept learning and reasoning that is goal-oriented, context-dependent, and analogical [14], challenging the objectivism (e.g., image classification by hyperplanes) embraced by traditional pattern recognition approaches [14, 15]. Therefore, they have been long known in the AI research community as an inspirational challenge for visual cognition. However, the original BPs are not amenable to today's data-driven visual recognition techniques, as this problem set is too small to train the state-of-the-art machine learning methods.

In this work, we introduce BONGARD-LOGO, a new benchmark for human-level visual concept learning and reasoning, directly inspired by the design principles behind the BPs. This new benchmark consists of 12,000 problem instances. The large scale of the benchmark makes it digestible by advanced machine learning methods in modern AI. To enable the scalable generation of Bongard-style problems, we develop a program-guided shape generation technique to produce human-interpretable visual pattern recognition problems in action-oriented LOGO language [16], where each visual pattern is generated by executing a sequence of program instructions. These problems are designed to capture the key properties of human cognition exhibited in original BPs, including 1) context-dependent perception, in which the same object may have fundamentally different interpretations given different contexts; 2) analogy-making perception, in which some meaningful concepts are traded off for other meaningful concepts; and 3) perception with a few samples but infinite vocabulary. These three properties together distinguish our benchmark from previous concept learning datasets that centered around standard recognition tasks [3, 7, 17, 18].

In our experiments, we formulate this task as a few-shot concept learning problem [4, 5] and tackle it with the state-of-the-art meta-learning [19–23] and abstract reasoning [17] algorithms. We find that all the models have significantly fallen short of human-level performances. This large performance gap between machine and human implies a failure of today's pattern recognition systems in capturing the core properties of human cognition. We perform both ablation studies and systematic analysis of the failure modes in different learning-based approaches. For example, the ability of learning to learn might be crucial for generalizing well to new concepts, as meta-learning methods largely outperform other approaches in our benchmark. Besides, each type of meta-learning methods has its preferred generalization tasks, according to their different performances on our test sets. Finally, we discuss potential research frontiers of building computational architectures for high-level visual cognition, driven by tackling the BONGARD-LOGO challenge.

## 2  BONGARD-LOGO Benchmark

Each puzzle in the original BP set is defined as follows: Given a set $\mathcal{A}$ of six images (positive examples) and another set $\mathcal{B}$ of six images (negative examples), the objective is to discover the rule (or concept) that the images in set $\mathcal{A}$ obey and images in set $\mathcal{B}$ violate. The solution to a BP is a logical rule stated in natural language that describes the visual concept presented in all images of set $\mathcal{A}$ but none of set $\mathcal{B}$. The original BPs consist of one hundred visual pattern recognition problems of black and white drawings. Through these carefully designed problems, M. M. Bongard aimed to demonstrate the key properties of human visual cognition capabilities and the challenges that machines have to overcome [11]. We highlight these key properties in detail in Section 2.2.

From a machine learning perspective, we can have multi-faceted interpretations of the BPs. Pioneer studies cast it as an inductive logic programming (ILP) problem [12] and a concept communication problem [13]. These two formulations typically require a significant amount of hand-engineering to define the logic rules or the language grammars, limiting their broad applicability. A more general and learning-oriented view of this problem is to cast it as a few-shot learning problem (as first mentioned in [24]), where the goal is to efficiently learn the concept from a handful of image examples. We are most interested in this formulation, as it requires the minimum amount of manual specification and likely leads to more generic approaches. In the next, we introduce our benchmark that inherits the properties of human cognition while being compatible with modern data-driven learning tools.

### 2.1  Benchmark Overview

We developed the BONGARD-LOGO benchmark that shares the same purposes as the original BPs for human-level visual concept learning and reasoning. Meanwhile, it contains a large quantity of 12,000 problems and transforms concept learning into a few-shot binary classification problem. Figure 1 shows some examples in BONGARD-LOGO. For example, in Figure 1c, set $\mathcal{A}$ contains six image patterns which are all convex shapes and, set $\mathcal{B}$ contains six image patterns which are all concave shapes. The task is to judge whether the pattern in the test image matches the concept (e.g., convex vs concave) induced by the set $\mathcal{A}$ or not. As the same concept might lead to vastly different patterns, a successful model must have the ability to identify the concept that distinguishes $\mathcal{A}$ and $\mathcal{B}$. The problems in BONGARD-LOGO belong to three types based on the concept categories:

**Free-form shape problems**   In the first type of problems, we have 3,600 *free-form shape* concepts, where each shape is composed of randomly sampled action strokes. The rationale behind these problems is that the concept of an image pattern can be *uniquely* characterized by the action program that generates it [3, 25]. Here the latent concept corresponds to the sequence of action stokes, such as straight lines, zigzagged arcs, etc. Among all 3,600 problems, each has a *unique* concept of strokes, and the number of strokes in each shape varies from two to nine and each image may have one or two shapes. As shown in Figure 1a, all images in the positive set form a one-shape concept and share the same sequence of six strokes that none of images in the negative set possesses. Note that some negative images may have subtle differences in stokes from the positive set, such as perturbing a stroke from `straight_line` into `zigzagged_line`, and please see Appendix D.1 for more examples. To solve these problems, the model may have to implicitly induce the underlying programs from shape patterns and examine if test images match the induced programs.

**Basic shape problems**   The other 4,000 problems in our benchmark are designed for the *basic shape* concepts, where the shapes are associated with 627 human-designed shape categories of large variation. The concept corresponds to recognizing one shape category or a composition of two shape categories presented in the shape patterns. Similar to the free-form shape problems, each problem has a unique basic shape concept. One important property of these problems is to test the analogy-making perception (see Section 2.2) where, for example, `zigzag` is an important feature in free-form shapes but a nuisance in basic shape problems. Figure 1b illustrate an instance of basic shape problems, where all six images in the positive set have the concept: a combination of "fan"-like shape and "trapezoid", while all six images in the negative set are other different combinations of two shapes. Note that positive images in Figure 1b may have zigzagged lines or lines formed by a set of circles (i.e., different stroke types), but they all share the same basic shape concept.

**Abstract shape problems**   The remaining 4,400 problems are aimed for the *abstract shape* concepts. Each shape is sampled from the same set of human-designed 627 shape categories. But in contrast to the previous type, these concepts correspond to more abstract attributes and their combinations. The

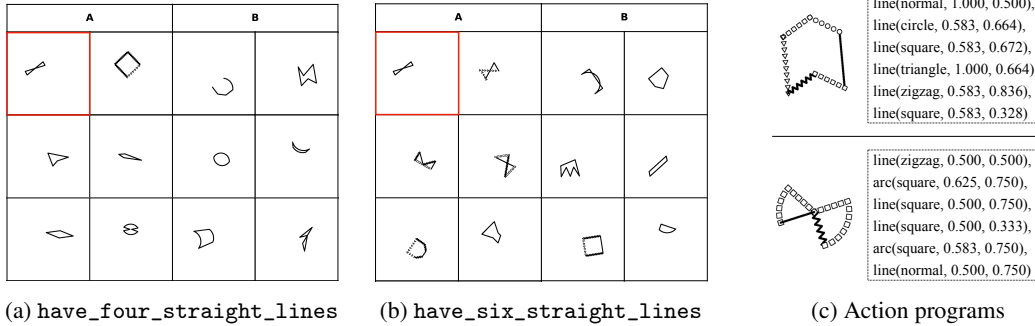

(a) `have_four_straight_lines`   (b) `have_six_straight_lines`   (c) Action programs

Figure 2: (a-b) An illustration of the context-dependent perception in BONGARD-LOGO, where (a) the concept is `have_four_straight_lines`, (b) the concept is `have_six_straight_lines`. The same shape pattern (highlighted in red) has fundamentally opposite interpretations depending on the context: Whether the line segments are seen as continuous when they intersect with each other. (c) Two exemplar shapes and the action programs that generate them procedurally, where each base action (`line` or `arc`) has three arguments (from left to right): *moving type*, *moving length* and *moving angle*. Note that there are five moving types, including `normal`, `zigzag`, `triangle`, `circle`, and `square`, each of which denotes how the line or arc is drawn. Also, both values of moving length and moving angle have been normalized into $[0, 1]$, respectively.

purpose of these problems is to test the ability of abstract concept discovery and reasoning. There are 25 abstract attributes in total, including `symmetric`, `convex`, `necked`, and so on. Each abstract attribute is shared by many different shapes while each shape is associated with multiple attributes. Due to the increased challenge of identifying the abstract concept, we randomly sample 20 different problems for each abstract shape concept. Figure 1c shows an example of abstract shape problems, where the underlying concept is `convex`. Similarly, the stroke types (i.e., zigzags or lines of circles, etc.) do not impact the convexity. Large variations in all convex and concave shapes ensure that the model does not simply memorize some finite shape templates but instead is forced to understand the gist of the `convex` concept.

## 2.2   Properties of the BONGARD-LOGO Problems

In the BONGARD-LOGO problems, we do not distinguish concepts by the shape size, orientation and position, or relative distance of two shapes. Furthermore, the Bongard-style reasoning needs the visual concept to be inferred from its context with a few examples. All these together demand a perception that is both rotation-invariant and scale-invariant, and more importantly, a new learning paradigm different from traditional image recognition methods. In the next, we mainly focus on three core characteristics of human cognition captured by BONGARD-LOGO, and use intuitive examples to explain them: 1) context-dependent perception, 2) analogy-making perception, and 3) perception with a few samples but of infinite vocabulary.

**Context-dependent perception**   The very same geometrical arrangement may have fundamentally different representations for each context on which it arises. For example, the underlying concept in the task of Figure 2a is `have_four_straight_lines`. Straight line segments must be seen as continuous, even when they intersect with another line segment. This does not happen, in the task of Figure 2b, where intersections have to split the straight lines to fulfill the underlying concept `have_six_straight_lines`. Another representative example is the hierarchy in concept learning. An equilateral triangle can either be interpreted as `equilateral_triangle` or as `convex`, which completely depends on what the remaining positive images and negative images are.

This property is compliant with the so-called "*one object, many views*" in human cognition [10, 26], where the existence and meaning of an object concept depend on the human understanding. Inspired by the original BPs, our benchmark contains a large number of the above examples that require context-dependent perception: $P(X, C)$, where $C$ is the context to which an object $X$ belongs. It does not make sense to infer the concept from $X$ alone, as there is a need for contextual information that lies outside of $X$. By contrast, most current pattern recognition models are implicitly *context-free*. For example, the image classification models take a single cat image and output a cat label, without referring to any other image as context. They assume that there exist objects in reality outside of any understanding, and the goal is to find the unique correct description [15]. Therefore, the perception

in the conventional context-free models is a one-to-one mapping from any object $X$ to its unique one representation $P(X)$. This may account for the inefficacy of current pattern recognition models, founded on the premise of context-free perception, in our benchmark.

**Analogy-making perception**    When humans make an analogy, we interpret an object in terms of another. In such a sense, representations can be traded off – we may interpret a zigzag as a straight line, or a set of triangles as an arc, or just about anything as another thing [15, 27]. When we trade off the zigzag for a straight line, we not only interpret it as a straight line, but also project onto it all the properties of a straight line, such as straightness. Though zigzags can never be straight, one could easily imagine a zigzagged line and a zigzagged arc [27], as shown in Figure 1.

Many problems in our benchmark require an analogy-making perception. Figure 1b shows an example where the underlying concept includes `trapezoid`, even if some trapezoids have no straight lines. Instead, they are a set of circles or zigzags that form a conceptual shape of `trapezoid`. Importantly, this may not be just an issue of noise or nuisance, because when we tackle the task, we trade off some concepts for other concepts. On one hand, we have circles or zigzags as a meaningful structure for the underlying concept of free-form problems. That is, we strictly distinguish a circle shape from a triangle shape, or a zigzagged line from a straight line in free-form problems. On the other hand, we trade off a set of triangles or zigzags for a `trapezoid` (Figure 1b) in the basic shape problems, and trade off a set of circles for a `convex` concept (Figure 1c) in the abstract attribute problems. A model with analogy-making perception will precisely know when a representation is crucial for the concept and when it has to be traded off for other concept.

**Perception with a few examples but infinite vocabulary**    Unlike standard few-shot image classification benchmarks [4, 20, 28], there is no finite set of categories to name or standard geometrical arrangements to describe in BONGARD-LOGO. Similar to original BPs, our problem set is not just about dealing with triangles, circles, squares, or other easily categorizable shape patterns. Particularly in many free-form shape problems, it would be a formidable task to explain their content to others in a context-free manner, if they have not seen the problems before. For example, Figure 1a shows a representative example of free-form problems, where we humans cannot provide the precise name of the free-form shape, as a concept, but we are still able to easily recognize the concepts by observing the strokes, and then make the right decisions on classifying novel test images [29]. As each free-form shape is an arbitrary composition of randomly sampled basic stroke structures. The space of all possible combinations makes the shape vocabulary size infinite.

The infinite vocabulary forbids a few-shot learner from memorizing geometrical arrangements in a dataset rather than developing an ability to conceptualize. This property is consistent with practical observations of human visual cognition that infer concepts from a few examples and generalize them to vastly different situations [3, 25].

## 2.3    Problem Generation with Action-Oriented Language

A distinctive feature of the original BPs is that a visual concept can be implicitly and concisely communicated by a comparison between two sets of image examples. It requires careful construction of the image sets $\mathcal{A}$ and $\mathcal{B}$ such that the concept can be identified with clarity. M. M. Bongard manually designed the original set of one hundred problems to convey the concepts he had in mind. However, this manual generation process is not scalable. We procedurally generated our shapes in the LOGO language [16], where a so-called "turtle" moves under a set of procedural action commands and its trajectory produces vector graphics. For each shape, the corresponding procedural action commands form its ground-truth *action program*.

The use of action programs to generate shapes has several benefits: First, with action programs, we can easily generate arbitrary shapes and precisely control the shape variation in a human-interpretable way. For example, in the generation of free-form shapes, we randomly perturb one command in the ground-truth action programs to construct a similar-looking and challenging negative set. Second, the ground-truth action programs provide useful supervision in guiding symbolic reasoning in the action space. It has a great potential of promoting future methods that use the symbolic stimuli, such as neuro-symbolic AI. Note that there exists a "*one-to-many*" relationship between the action program and shape pattern: an action program *uniquely* determines a shape pattern while the same shape pattern may have *multiple* correct action programs that can generate it. Please see Appendix C for our preliminary results of incorporating symbolic information into neural networks for better performance. More importantly, although our benchmark has an infinite vocabulary of shape patterns,

the vocabulary of base actions are of relatively small size, making the concepts compositional in the action space and much easier to generate.

To simplify the process, we use only two classes of base actions: `line` and `arc`. As each base action has three arguments: *moving type*, *moving length* and *moving angle*, an action is depicted by a function: `[action_name]([moving_type], [moving_length], [moving_angle])`, as shown in Figure 2c. Specifically for the above arguments, there are five moving types, including `normal`, `zigzag`, `triangle`, `circle`, and `square`. For instance, `normal` means the line or arc is perfectly straight or curved, `zigzag` means the line or arc is of the zigzagged-style and `triangle` means the line or arc is formed by a set of triangles. Besides, both moving length and moving angle are normalized into the range of $[0, 1]$, where 0 and 1 correspond to the minimal and maximal values, respectively. Depending on varying lengths of action programs, different combinations of these base actions form visually distinct shapes. Our benchmark contains different types of shapes, as discussed in Section 2.1, each of which requires a separate procedure in the LOGO language to generate, and we leave generation details in Appendix A.

We have open-sourced the procedural generation code and data of BONGARD-LOGO in the following GitHub repository: `https://github.com/NVlabs/Bongard-LOGO`.

## 3 Experiments

### 3.1 Methods

We consider several state-of-the-art (SOTA) approaches, and test how they behave in BONGARD-LOGO, which demands human cognition abilities. First, as each task in BONGARD-LOGO can be cast as a *two-way six-shot* few-shot classification problem, where meta-learning has been a standard framework [30, 4], we first introduce the following meta-learning methods, with each being SOTA in different (i.e., memory-based, metric-based, optimization-based) meta-learning categories: 1) *SNAIL* [19], a memory-based method; 2) *ProtoNet* [20], a metric-based method; 3) *MetaOptNet* [21] and *ANIL* [22], two optimization-based methods. As the Meta-Baseline [23] is a new competitive baseline in many few-shot classification tasks, we consider its two variants: 1) *Meta-Baseline-SC*, where we meta-train the model from scratch, and 2) *Meta-Baseline-MoCo*, where first use an unsupervised contrastive learning method – MoCo [31] to pre-train the backbone network and then apply meta-training. Besides, we consider two non-meta-learning baselines for comparison. One is called *WReN-Bongard*, a variant of WReN [17] that was originally designed to encourage reasoning in the Raven-style Progressive Matrices (RPMs) [32]. Another one is a convolutional neural network (CNN) baseline, by casting the task into a conventional binary image classification problem, which we call *CNN-Baseline*. Please see Appendix B for the detailed descriptions of different models.

### 3.2 Benchmarking on BONGARD-LOGO

We split the 12,000 problems in BONGARD-LOGO into the disjoint train/validation/test sets, consisting of 9300, 900, and 1800 problems respectively. In the training set and validation set, we uniformly sample problems from three problem types in Section 2.1. To dissect the performances of the models in different problem types and levels of generalization, we use the following four test set splits:

**Free-form shape test set** This test set (FF) includes 600 free-form shape problems. To evaluate the capability of these models in *extrapolation* towards more complex free-form shape concepts than trained concepts, the shape patterns in this test set are generated with "one longer" action programs than the longest programs used for generating the training set. Note that we consider the test set of "one longer" action programs as a basic case for extrapolation. Since this task has been shown difficult for current methods (as shown in Table 1), we did not include more challenging setups.

**Basic shape test set** This test set (BA) includes 480 basic shape problems. We randomly sample 480 basic shape problems in which the concept corresponds to recognizing a composition of two shapes. As each basic shape concept only appears once in our benchmark, we ensure that the basic shape test set shares no common concept with the training set. This test set is designed to examine a model's ability to generalize towards novel *composition* of basic shape concepts.

**Combinatorial abstract shape test set** This test set (CM) includes 400 abstract shape problems. We randomly sample 20 novel pairwise combinations of two abstract attributes, each with 20 problems. All the single attributes in this test set have been observed individually in the training set. However,

| Methods | Train Acc | Test Acc (FF) | Test Acc (BA) | Test Acc (CM) | Test Acc (NV) |
|---|---|---|---|---|---|
| SNAIL [19] | $59.2 \pm 1.0$ | $56.3 \pm 3.5$ | $60.2 \pm 3.6$ | $60.1 \pm 3.1$ | $61.3 \pm 0.8$ |
| ProtoNet [20] | $73.3 \pm 0.2$ | $64.6 \pm 0.9$ | $72.4 \pm 0.8$ | $62.4 \pm 1.3$ | $\mathbf{65.4 \pm 1.2}$ |
| MetaOptNet [21] | $75.9 \pm 0.4$ | $60.3 \pm 0.6$ | $71.7 \pm 2.5$ | $61.7 \pm 1.1$ | $63.3 \pm 1.9$ |
| ANIL [22] | $69.7 \pm 0.9$ | $56.6 \pm 1.0$ | $59.0 \pm 2.0$ | $59.6 \pm 1.3$ | $61.0 \pm 1.5$ |
| Meta-Baseline-SC [23] | $75.4 \pm 1.0$ | $\mathbf{66.3 \pm 0.6}$ | $\mathbf{73.3 \pm 1.3}$ | $63.5 \pm 0.3$ | $63.9 \pm 0.8$ |
| Meta-Baseline-MoCo [23] | $\mathbf{81.2 \pm 0.1}$ | $65.9 \pm 1.4$ | $72.2 \pm 0.8$ | $\mathbf{63.9 \pm 0.8}$ | $64.7 \pm 0.3$ |
| WReN-Bongard [17] | $78.7 \pm 0.7$ | $50.1 \pm 0.1$ | $50.9 \pm 0.5$ | $53.8 \pm 1.0$ | $54.3 \pm 0.6$ |
| CNN-Baseline | $61.4 \pm 0.8$ | $51.9 \pm 0.5$ | $56.6 \pm 2.9$ | $53.6 \pm 2.0$ | $57.6 \pm 0.7$ |
| Human (Expert) | - | $92.1 \pm 7.0$ | $99.3 \pm 1.9$ | $90.7 \pm 6.1$ | |
| Human (Amateur) | - | $88.0 \pm 7.6$ | $90.0 \pm 11.7$ | $71.0 \pm 9.6$ | |

Table 1: Model performance versus human performance in BONGARD-LOGO. We report the test accuracy (%) on different dataset splits, including free-form shape test set (FF), basic shape test set (BA), combinatorial abstract shape test set (CM), and novel abstract shape test set (NV). Note that for human evaluation, we report the separate results across two groups of human subjects: *Human (Expert)* who well understand and carefully follow the instructions, and *Human (Amateur)* who quickly skim the instructions or do not follow them at all. The chance performance is 50%.

the 20 novel combinations are exclusive in the test set. The rationale behind this test set is that understanding the abstract shape concepts requires a model's *abstraction* ability, as it is challenging to conceptualize abstract meanings from large shape variations.

**Novel abstract shape test set**    This test set (NV) includes 320 abstract shape problems. Our goal here is to evaluate the ability of a model on the *discovery* of new abstract concepts. Different from the construction of the combinatorial abstract shape test set (CM), we hold out one attribute and all its combinations with other attributes from the training set. All problems related to the held-out attribute are exclusive in this test set. Specifically, we choose `"have_eight_straight_lines"` as the held-out attribute, since it presumably requires minimal effort for the model to extrapolate given that other similar `"have_[xxx]_straight_lines"` attributes already exist in the training set.

### 3.3   Quantitative Results

We report the test accuracy (Acc) of different methods on each of the four test sets respectively, and compare the results to the human performance in Table 1. We put the experiment setup for training these methods to Appendix C, and the results are averaged across three different runs. To show the human performance on our benchmark, we choose 12 human subjects to test on randomly sampled 20 problems from each test set. Note that for human evaluation, we do not differentiate test set (CM) and test set (NV) and thus report only one score on them, as humans essentially perform the same kind of new abstract discovery on both of the two test sets. To familiarize the human subjects with BONGARD-LOGO problems, we describe each problem type and provide detailed instructions on how to solve these problems. It normally takes 30-60 minutes for human subjects to fully digest the instructions. Depending on the total time that a human subject spends on digesting instructions and completing all tasks, we have split 12 human subjects into two evenly distributed groups: *Human (Expert)* who well understand and carefully follow the instructions, and *Human (Amateur)* who quickly skim the instructions or do not follow them at all.

**Performance analysis**    In Table 1, we can see that there exists a significant gap between the Human (Expert) performance and the best model performance across all different test sets. Specifically, Human (Expert) can easily achieve nearly perfect performances (>99% test accuracy) on the basic shape (BA) test set, while the best performing models only achieve around 70% accuracy. On the free-form shape (FF), combinatorial abstract shape (CM) and novel abstract shape (NV) test sets where the existence of infinite vocabulary or abstract attributes makes these problems more challenging, Human (Expert) can still get high performances (>90% test accuracy) while all the models only have around or less than 65% accuracy. The considerable gap between these SOTA learning approaches and human performance implies these models fail to capture the core human cognition properties, as mentioned in Section 2.2.

Comparing Human (Expert) and Human (Amateur), we can see Human (Expert) always achieve better test accuracies with lower variances. This advantage becomes much more significant in the abstract shape problems, confirming different levels of understanding of these abstract concepts among human subjects. Comparing different types of methods, meta-learning methods that have been specifically

designed for few-shot classification have a greater potential to address the BONGARD-LOGO tasks than the non-meta-learning baselines (i.e., WReN-Bongard and CNN-Baseline). In Table 1, we can see the better test accuracies of most meta-learning models across all the test sets, compared with the non-meta-learning models. Interestingly, WReN, the model for solving RPMs [17], suffers from the severest overfitting issues, where its training accuracy is around 78% but its test accuracies are only marginally better than random guess (50%). This manifests that the ability of learning to learn is crucial for generalizing well to new concepts.

Furthermore, we perform an ablation study on BONGARD-LOGO, where we train and evaluate on the subset of 12,000 free-form shape problems in the same way as before. As the properties of *context-dependent perception* and *analogy-making perception* are not strongly presented in these free-form problems, concept learning on this subset has a closer resemblance to standard few-shot visual recognition problems [4, 33]. We thus expect a visible improvement in their performances. This is confirmed by the results in Table 2 in Appendix C, where almost all the methods achieve better training and test performances. Specifically, the best training accuracy of methods increases from 81.2% to 96.4%, and the best test accuracy (FF) of methods increases from 66.3% to 74.5%. However, there still exists a large gap between the model and human performance on free-form shape problems alone (74.5% vs. 92.1%). It implies the property of *few-shot learning with infinite vocabulary* has already been challenging for these methods.

We then compare different meta-learning methods and diagnose their respective failure modes. First, Meta-Baseline-MoCo performs best on the training set and also achieves competitive or better results on most of the test sets. It confirms the observations in prior work that good representations play an integral role in the effectiveness of meta-learning [23, 28]. Between the two Meta-Baselines, we can see that the MoCo pre-training is more effective in improving the training accuracy but the improvements become marginal on the test sets. This implies that the SOTA unsupervised pre-training methods improve the overall performance while still facing severe overfitting issues. Second, we perform another ablation study on model sizes. The results in Figure 5 in Appendix C show that the generalization performances of different models generally get worse as model size decreases.

Besides, most meta-learning models perform much better on the basic shape problems than on the abstract shape problems. This phenomenon is consistent with the human experience, as it takes a much longer time for humans to finish a latter task. How to improve the combinatorial generalization or extrapolation of abstract concepts in our benchmark remains a challenge for these models. We also observe that each type of meta-learning methods has its preferred generalization tasks, according to their different behaviors on test sets. For example, the memory-based method (i.e., SNAIL) seems to try its best to perform well on abstract shape problems by sacrificing its performance in the free-form shape and basic shape problems. The metric-based method (i.e., ProtoNet) performs similarly to Meta-Baselines and better than memory-based and optimization-based methods. Lastly, the optimization-based methods (i.e., MetaOptNet and ANIL) tend to have a larger gap between training and test accuracies. These distinguishable behaviors on our benchmark provide a way of quantifying the potential advantages of a meta-learning method in different use cases.

## 4  Related Work

**Few-shot learning and meta-learning**    The goal of few-shot learning is to learn a new task (e.g., recognizing new object categories) from a small amount of training data. Pioneer works have approached it with Bayesian inference [33] and metric learning [34]. A rising trend is to formulate few-shot learning as meta-learning [4, 5]. These methods can be categorized into three families: 1) memory-based methods, e.g., a variant of MANN [30] and SNAIL [19], 2) metric-based methods, e.g., Matching Networks [35] and ProtoNet [20], and 3) optimization-based methods, e.g., MAML [5], MetaOptNet [21] and ANIL [22]. Recent work [23, 28] has achieved competitive or even better performances on few-shot image recognition benchmarks [3, 36] with a simple pre-training baseline than advanced meta-learning algorithms. It also sheds light on a rethinking of few-shot image classification benchmarks and the associated role of meta-learning algorithms.

**Concept learning and Bongard problem solvers**    Concept learning methods concern about transforming sensory observations [37] and experiences [38] into abstract concepts, which serve as the building blocks of understanding and reasoning. A rich and structured representation of concepts is programs [3, 25] that model the generative process of observed data in an analysis-by-synthesis framework. Program-based (or rule-based) methods have been applied to tackle BPs [12, 39, 13].

RF4 [12] is an ILP system where the images are hand-coded into logical formulas. Phaeaco [39] is an exemplar BP solver that implements the FARG concept architecture [10]. [13] proposes to translate visual features into a formal language with which the solver applies Bayesian inference for concept induction. However, these methods require a substantial amount of domain knowledge and do not offer complete solutions to original BPs, making them infeasible for our benchmark which consists of tens of thousands of problems. In contrast, [24] is the first work to show the usefulness of deep learning methods for solving BPs by using a pre-trained feature extracting neural network. All these results motivate the development of hybrid concept learning systems that integrate deep learning with symbolic reasoning, which have attracted much attention recently [40, 41]. The hybrid systems can offer great potential in combining the representational power of neural networks with the generalization power of symbolic operations for tackling our benchmark.

**Abstract reasoning benchmarks**    In addition to popular machine perception benchmarks that focus on categorization and detection [1, 2], there have been previous efforts in creating new benchmarks for abstract reasoning, including compositional visual question answering [7], physical reasoning [42–44], mathematics problems [45], and general artificial intelligence [46]. Most relevant to our benchmark, the Raven-style Progressive Matrices (RPMs) [47], a well-known human IQ, have inspired researchers to design new abstract reasoning benchmarks [17, 48]. Our benchmark is complementary to RPMs: RPMs focus on relational concepts (such as progression, XOR, etc.) while BONGARD-LOGO problems focus on object concepts (such as stroke types, abstract attributes, etc.). In RPMs, the relational concepts come from a small set of five relations [17]. In our benchmark, the object concepts can vary arbitrarily with procedural generation. Recently, [49] has proposed V-RPM that extends RPMs to real images. As V-PROM is a real image version of RPMs, it differs from our benchmark in the similar ways described above. Therefore, in contrast to RPMs where automated pattern recognition models have achieved competitive performances [50, 17], our BONGARD-LOGO benchmark has posed a greater challenge to current models due to its fundamental shift towards more human-like (i.e., context-based, analogy-making, few-shot with infinite vocabulary) visual cognition.

## 5    Discussion and Future Work

We introduced a new visual cognition benchmark that emphasizes concept learning and reasoning. Our benchmark, named BONGARD-LOGO, is inspired by the original BPs [11] that were carefully designed in the 1960s for demonstrating the chasms between human visual cognition and computerized pattern recognition. In a similar vein as the original one hundred BPs, our benchmark aims for a new form of human-like perception that is context-dependent, analogical, and few-shot of infinite vocabulary. To fuel research towards new computational architectures that give rise to such human-like perception, we develop a program-guided problem generation technique that enables us to produce a large-scale dataset of 12K human-interpretable problems, making it digestible by data-driven learning methods to date. Our empirical evaluation of the state-of-the-art visual recognition methods has indicated a considerable gap between machine and human performance on this benchmark. It opens the door to many research questions: What types of inductive biases would benefit concept learning models? Are deep neural networks the ultimate key to human-like visual cognition? How is the Bongard-style visual reasoning connected to semantics and pragmatics in natural language?

Prior attempts on tackling the orginal BPs with symbolic reasoning [12] and program induction [13] have also been far away from solving BONGARD-LOGO. However, along with the inherent nature of compositionality and abstraction in BPs, they have supplied a great amount of insight on the path forward. Therefore, one promising direction is to develop computational approaches that integrate neural representations with symbolic operations in a hybrid system [8, 40], that acquires and reasons about abstract knowledge in a prolonged process [51], that establishes a tighter connection between visual perception and the high-level cognitive process [14, 3]. We invite the broad research community to explore these open questions together with us for future work.

## Broader Impact

Our work created a new visual concept learning benchmark inspired by the Bongard problems. Our preliminary evaluations have illustrated a considerable gap between human cognition and machine recognition, highlighting the shortcomings of existing pattern recognition methods. In Machine Learning and Computer Vision, we have witnessed the integral role of standardized benchmarks [1, 2] in promoting the development of new AI algorithms. We would encourage future work to develop

new visual cognition algorithms towards human-level visual concept learning and reasoning. We envision our benchmark to serve as a driving force for research on context-dependent and analogical perception beyond standard visual recognition. We believe that endowing machine perception with the abilities to learn and reason in a human-like way is an essential step towards building robust and reliable AI systems in the wild. It could potentially lead to more human-interpretable AI systems and address concerns about ethics and fairness arising from today's data-driven learning systems that inherit or augment the biases in training data. A potential risk of our new benchmark is that it might skew research towards highly customized methods without much applicability for more general concept learning and reasoning. We encourage researchers to develop new algorithms for our benchmark from the first principles and avoid highly customized solutions, to retain the generality and broad applicability of the resultant algorithms.

## Acknowledgments and Disclosure of Funding

We thank the anonymous reviewers for useful comments. We also thank all the human subjects for participating in our BONGARD-LOGO human study, and the entire AIALGO team at NVIDIA for their valuable feedback. WN conducted this research during an internship at NVIDIA. WN and ABP were supported by IARPA via DoI/IBC contract D16PC00003 and NSF NeuroNex grant DBI-1707400.

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
