[Supplementary Material]

# Appendix

## A More Details on Problem Generation with LOGO

As described in Section 2.1, three types of BONGARD-LOGO problems consist of two types of shapes. Free-form shape problems only contain *free-form shapes*, while both basic shape problems and abstract shape problems are composed of samples from 627 *human-designed shapes*. Since we have discussed how to use rendered shapes (i.e., *free-form* and *human-designed*) to form different types of problems, we will next talk about the process of generating each type of shapes.

**Free-form shapes** A free-form shape is generated in the following steps: 1) Decide what length of action programs a shape has. 2) Randomly and independently sample each base action and its corresponding three moving arguments (i.e., *moving type*, *moving length* and *moving angle*) in sequence, resulting in a tentative action program. 3) Execute the above tentative action program in the shape renderer (i.e., turtle graphics) to generate the shape. Sometimes, the strokes of the rendered shape are heavily overlapped with each other, making the shape difficult to recognize. If it has happened, we go back to step 2) and repeat the process until either there is no heavy overlapping or the maximum number of steps has reached. Note that for the free-form shape generation, the positive and negative images would be easily indistinguishable if the moving length and angle of each action have continuous values. Therefore, we further constrain the moving length and angle to be only sampled from a set of well-separated discrete values.

**Human-designed shapes** Among 627 human-designed shapes, each one is paired with a ground-truth action stokes. All these shapes are stored in a dictionary with a key-value pair: (*shape name*, *action stokes*). The only difference between *action program* and *action stokes* is that the latter does not specify the *moving type*, as both basic shape and abstract shape problems treat it as one of the nuisances. Therefore, a human-designed shape is generated in the following steps: 1) Randomly sample a shape name from a predetermined subset of 627 shape names. Note that for basic shape problems, the predetermined subset is depicted by shape categories (such as triangles, squares, etc.), while for abstract shape problems, the predetermined subset is depicted by abstract attributes (such as convex, symmetric, etc.). 2) Infer its corresponding *action stokes* by looking up the dictionary and add randomly sampled *moving type* to each action, resulting in the final action program. 3) Execute the above action program in the shape renderer (i.e., turtle graphics) to generate the shape.

During the generation process of both two types of shapes, we will randomize the initial starting point and initial moving angle of each shape, and the size of unit length, such that shape position, shape orientation and shape size are all considered as nuisances. It demands a perception that is both rotation-invariant and scale-invariant.

## B More Details on Methods

In this section, we provide more details on state-of-the-art (SOTA) meta-learning approaches and other strong baselines. As BONGARD-LOGO problems can be cast as a *two-way six-shot* few-shot classification problem, where meta-learning has been a standard framework [30, 4], we first discuss different SOTA meta-learning methods:

**SNAIL [19]** SNAIL is a memory-based meta-learning method. It proposed a class of simple and generic meta-learner architectures that use a novel combination of temporal convolutions and soft attention, with the former to aggregate information from past experience and the latter to pinpoint specific pieces of information.

**ProtoNet [20]** ProtoNet is a metric-based meta-learning method. It proposed a simple method called prototypical networks based on the idea that we can represent each class by the mean of its examples in a representation space learned by a neural network. In the learned metric space, classification can be performed by computing distances to prototype representations of each class.

**MetaOptNet [21]** MetaOptNet is an optimization-based meta-learning method. It proposed to learn the feature representation that can generalize well for a linear support vector machine (SVM) classifier.

Figure 3: WReN-Bongard, where the key idea is to use relation network to form representations of pair-wise relation between each context (i.e., positive or negative) image and a given test image, and between context images themselves. Note that the 'logit' will pass into a sigmoid function for binary classification.

Figure 4: CNN-Baseline, where we first stack the given test image and all six positive (or all six negative images) to form a "stacked image" with input seven channels. The two stacked images pass into ResNet to extract the respective features for obtaining the final logit. Similarly, the 'logit' will pass into a sigmoid function for binary classification.

By exploiting the dual formulation and KKT conditions, MetaOptNet enabled a computational and memory efficient meta-learning with higher embedding dimensions for improved performance.

**ANIL [22]** ANIL is an optimization-based meta-learning method. It is a simplification of MAML [5] where we remove the inner loop for all but the (task-specific) head of the underlying neural network. ANIL matched MAML's performance on several few-shot image classification benchmarks with significantly improved computational and memory efficiency.

**Meta-Baseline-SC & -MoCo** Meta-Baseline [23] is a new competitive baseline designed to investigate the role of feature representations in few-shot learning. It proposed to pre-train a classifier on all base classes and then meta-learn with a nearest-centroid based few-shot classification algorithm. Because there is no base class in BONGARD-LOGO, a supervised image classification pre-training is infeasible. Instead, we introduced two variants: 1) Meta-Baseline-SC, where we meta-train the Meta-Baseline from scratch, and 2) Meta-Baseline-MoCo, where we first use an unsupervised contrastive learning method MoCo [31] to pre-train the backbone model and then apply meta-training.

Besides, we consider two strong non-meta-learning baselines for comparison.

**WReN-Bongard** WReN [17] is a new architecture, designed to encourage reasoning, by solving visual IQ tests, the Raven-style Progressive Matrices (RPMs) [32]. It has achieved strong performances on the RPMs. The idea of WReN is to form representations of pair-wise relation between each context and a given choice candidate, and between contexts themselves. We apply its relation network module to our problems. To do so, we develop a variant of WReN, named WReN-Bongard, by adapting WReN to Bongard problems. Figure 3 shows the architecture of WReN-Bongard.

**CNN-Baseline** CNN is a standard deep learning baseline of image classification without meta-learning or relation reasoning. The idea of CNN-Baseline is to stack the test image in each problem alongside all six positive images and all six negative images, respectively, to form two "stacked images" with seven input channels. This way, we convert the few-shot learning problem to a conventional binary image classification problem. Figure 4 shows the architecture of CNN-Baseline.

# C  More Experiment Details

**Experiment setup**    Each image in BONGARD-LOGO is grey-scale with resolution $512 \times 512$. We train each model in Section 3.1 on the training set for 100 epochs. For a fair comparison, we use ResNet-15 (where feature map sizes are 32-64-128-256-512) with output feature dimension 128 as the backbone network in all the above methods. We use the SGD optimizer with momentum 0.9 and learning rate 0.001 with weight decay 5e-4. For all meta-learning models, we use a batch size eight on 8 GPUs, namely that each training batch contains eight Bongard problems for the loss computation. For CNN-Baseline, we use a batch size 32 on 8 GPUs. For each run of all the considered models on 8 NVIDIA V100 GPUs, it takes less than 12 hours to get the results. The other hyperparameters are kept the same with the default implementations as the respective original work.

**Ablation study on BONGARD-LOGO**    Here we create a variant of BONGARD-LOGO, where we only include 12,000 free-form shape problems. As the properties of *context-dependent perception* and *analogy-making perception* are not presented any more, concept learning on this variant has a closer resemblance to standard few-shot visual recognition problems [4, 33]. Thus, we expect a large improvement in the performances of these methods. The results of model evaluation and human study in this variant of BONGARD-LOGO, are shown in Table 2. We can see that almost all the considered methods achieve better training and test performances. Specifically, the best training accuracy of methods increases from 81.2% to 96.4%, and the best test accuracy (FF) of methods increases from 66.3% to 74.5%. However, there still exists a large gap between the model and human performance on free-form shape problems alone. It implies the property of *few-shot learning with infinite vocabulary* has already been challenging for current methods. Another observation is that WReN-Bongard outperforms most meta-learning methods on this variant of BONGARD-LOGO, demonstrating its potential as a strong baseline for concept learning and reasoning in simpler cases.

| Methods | Train Acc | Test Acc (FF) |
|---|---|---|
| SNAIL [19] | $74.4 \pm 2.5$ | $65.2 \pm 2.0$ |
| ProtoNet [20] | $90.5 \pm 0.6$ | $68.5 \pm 0.7$ |
| MetaOptNet [21] | $91.5 \pm 0.7$ | $66.9 \pm 0.5$ |
| ANIL [22] | $77.8 \pm 0.6$ | $63.2 \pm 0.7$ |
| Meta-Baseline-SC [23] | $92.4 \pm 0.3$ | $70.8 \pm 0.2$ |
| Meta-Baseline-MoCo [23] | $95.8 \pm 0.3$ | $72.5 \pm 0.5$ |
| WReN-Bongard [17] | $\mathbf{96.4 \pm 0.8}$ | $\mathbf{74.5 \pm 4.0}$ |
| CNN-Baseline | $79.9 \pm 6.2$ | $49.8 \pm 1.0$ |
| Human (Expert) | - | $92.1 \pm 7.0$ |
| Human (Amateur) | - | $88.0 \pm 7.6$ |

Table 2: Model performance versus human performance in a variant of BONGARD-LOGO which only includes 12,000 free-form shape problems. We report the training and test accuracy (%) on the free-form shape test set (FF). Note that for human evaluation, we report the separate results across two groups of human subjects: *Human (Expert)* who well understand and carefully follow the instructions, and *Human (Amateur)* who quickly skim the instructions or do not follow them at all. The chance performance is 50%.

**Capability of the CNN backbone on our visual stimuli**    Since the images in our benchmark are black-and-white drawings with fine lines, one may have a concern about whether the "visual recognition" part requires different capabilities from what the state-of-the-art CNN models designed for natural images. To evaluate the capability of the CNN backbone on our visual stimuli, regardless of the concept learning and reasoning aspects, we train a standard supervised recognition task with a training set of the visual inputs sampled from our benchmark. In particular, we train two separate binary attribute classifiers for `convex` and `have_two_parts`, respectively, using the same ResNet-15 backbone as in the paper. With each dataset of randomly sampled 14K positive and 14K negative samples, the model achieved the near-perfect train/test accuracies 98.7%/97.0% on `convex` and 98.0%/97.1% on `have_two_parts`. These results demonstrate that the CNN backbone is capable of processing the visual stimuli of our BONGARD-LOGO tasks.

**Ablation study on model sizes**    To show how model sizes affect the concept learning performance, we vary the model size by dividing each layer size in the ResNet-15 backbone by a reduction factor

$\alpha$, where $\alpha$ is a divisor of the number of parameters. Figure 5 illustrates the training and test results of two top-performing models: ProtoNet and Meta-Baseline-SC. Note that the model size decreases as $\alpha$ increases. We see that both training accuracies decrease consistently with smaller model sizes. On the test sets, the generalization performances also generally get worse as model size decreases, but results slightly vary across different models. ProtoNet is more sensitive to model size than Meta-Baseline-SC: (a) All the test accuracies of ProtoNet tend to decrease with smaller model sizes, while (b) test accuracies of Meta-Baseline-SC mostly remain robust to various model sizes (except for the extreme case of $\alpha = 16$).

Figure 5: Performance of Meta-Baseline-SC and ProtoNet with different model sizes, controlled by a reduction factor $\alpha$ (i.e., the model size is smaller with a larger $\alpha$).

**Incorporating Symbolic Information for Better Performance**   Since there is a significant gap between model and human performance in our benchmark, as shown in Table 1, we have discussed the potential of neuro-symbolic approaches to close the gap in Section 5. Here we move one step forward towards a hybrid model and show some preliminary results of incorporating the symbolic information into neural networks. The basic idea is to replace the MoCo pre-training in Meta-Baseline-MoCo with the pre-training of a program synthesis task. We call the new model *Meta-Baseline-PS*, which stands for Meta-Baseline based on program synthesis.

Figure 6 shows the model illustration of (a) Meta-Baseline-PS and (b) one of its components – the *action decoder*. In Meta-Baseline-PS, we first pass the input images into the CNN backbone to extract image features, which are the input of two following branches: 1) the program synthesis module where we use LSTM to convert image features into action features and then use an action decoder to synthesize the action programs; 2) the meta-learner which we use to solve the BONGARD-LOGO problems. The training of Meta-Baseline-PS is composed of two stages, i.e., we first pre-train the program synthesis module to extract the symbolic-aware image feature and then fine-tune it by training the meta-learner.

| Methods | Train Acc | Test Acc (FF) | Test Acc (BA) | Test Acc (CM) | Test Acc (NV) |
|---|---|---|---|---|---|
| SNAIL [19] | $59.2 \pm 1.0$ | $56.3 \pm 3.5$ | $60.2 \pm 3.6$ | $60.1 \pm 3.1$ | $61.3 \pm 0.8$ |
| ProtoNet [20] | $73.3 \pm 0.2$ | $64.6 \pm 0.9$ | $72.4 \pm 0.8$ | $62.4 \pm 1.3$ | $\mathbf{65.4 \pm 1.2}$ |
| MetaOptNet [21] | $75.9 \pm 0.4$ | $60.3 \pm 0.6$ | $71.7 \pm 2.5$ | $61.7 \pm 1.1$ | $63.3 \pm 1.9$ |
| ANIL [22] | $69.7 \pm 0.9$ | $56.6 \pm 1.0$ | $59.0 \pm 2.0$ | $59.6 \pm 1.3$ | $61.0 \pm 1.5$ |
| Meta-Baseline-SC [23] | $75.4 \pm 1.0$ | $\mathbf{66.3 \pm 0.6}$ | $\mathbf{73.3 \pm 1.3}$ | $63.5 \pm 0.3$ | $63.9 \pm 0.8$ |
| Meta-Baseline-MoCo [23] | $\mathbf{81.2 \pm 0.1}$ | $65.9 \pm 1.4$ | $72.2 \pm 0.8$ | $\mathbf{63.9 \pm 0.8}$ | $64.7 \pm 0.3$ |
| WReN-Bongard [17] | $78.7 \pm 0.7$ | $50.1 \pm 0.1$ | $50.9 \pm 0.5$ | $53.8 \pm 1.0$ | $54.3 \pm 0.6$ |
| CNN-Baseline | $61.4 \pm 0.8$ | $51.9 \pm 0.5$ | $56.6 \pm 2.9$ | $53.6 \pm 2.0$ | $57.6 \pm 0.7$ |
| Meta-Baseline-PS | $\mathbf{85.2 \pm 1.0}$ | $\mathbf{68.2 \pm 0.3}$ | $\mathbf{75.7 \pm 1.5}$ | $\mathbf{67.4 \pm 0.3}$ | $\mathbf{71.5 \pm 0.5}$ |
| Human (Expert) | - | $92.1 \pm 7.0$ | $99.3 \pm 1.9$ | $90.7 \pm 6.1$ | |
| Human (Amateur) | - | $88.0 \pm 7.6$ | $90.0 \pm 11.7$ | $71.0 \pm 9.6$ | |

Table 3: Model performance versus human performance in BONGARD-LOGO. We report the test accuracy (%) on different dataset splits, including free-form shape test set (FF), basic shape test set (BA), combinatorial abstract shape test set (CM), and novel abstract shape test set (NV). Note that for human evaluation, we report the separate results across two groups of human subjects: *Human (Expert)* who well understand and carefully follow the instructions, and *Human (Amateur)* who quickly skim the instructions or do not follow them at all. Note that Meta-Baseline-PS means the Meta-Baseline based on program synthesis, which incorporates symbolic information to solve the task. The chance performance is 50%.

**Program Synthesis**

(a) Meta-Baseline-PS

(b) Action decoder

Figure 6: (a) Meta-Baseline-PS, where we first pass the input images into the CNN backbone to extract image features, which are the input of two following branches: 1) the program synthesis module where we use LSTM to convert image features into action features and then use an action decoder to synthesize the action programs; 2) the meta-learner which we use to solve the BONGARD-LOGO problems. (b) the *action decoder*, the architecture for inferring action command from the action feature, where 'MLP', 'MDN' and 'MoG' stand for Multi-Layer Perceptron, Mixture Density Network and Mixture of Gaussians, respectively.

In the action decoder, the action feature from LSTM is passed into each module to sequentially predict each token in the action command, as introduced in Section 2.3. Because the values of *action name* and *moving type* are discrete, we simply use MLP and softmax to predict their values. In contrary, the values of *moving length* and *moving angle* are continuous. To learn prediction with several distinct future possibilities [52], we apply the Mixture Density Network (MDN) [53] together with the Mixture of Gaussians (MoG) parameterizations to predict their values. Experiments have shown MDN achieves better performance than the vanilla L2-norm informed prediction.

Table 3 shows the training and test of Meta-Baseline-PS on BONGARD-LOGO, along with the results of SOTA baseline approaches and human subjects. We can see that Meta-Baseline-PS largely outperforms all the SOTA baselines. It demonstrates that incorporating symbolic information into neural networks improves the overall performance, confirming the great potential of neuro-symbolic methods on tackling the BONGARD-LOGO benchmark. However, by realizing that there is still a large gap between Meta-Baseline-PS and human performance, we leave the exploration of more advanced neuro-symbolic approaches to tackle the challenge of our benchmark, as the future work.

# D   More Examples in BONGARD-LOGO

We provide more examples from three types of problems, respectively, where each concept in any problem could be about one shape or a combination of two shapes.

## D.1    More Examples of Free-Form Shape Problems

(a) # of actions: [7]

(b) # of actions: [3, 5]

(c) # of actions: [8]

(d) # of actions: [4, 4]

Figure 7: More examples of free-form shape problems, where (a) the shape is generated by five action strokes, (b) the two shapes are generated by three and five action strokes, respectively, (c) the shape is generated by eight action strokes, (d) the two shapes are generated by four and four action strokes, respectively. In each problem, set $\mathcal{A}$ contains six images that satisfy the concept and set $\mathcal{B}$ contains six images that violate the concept. We also show two test images (left: positive, right: negative) in the binary classification problems. In free-form shape problems, the task is about discovering the concept of base strokes in a shape, which may differ in inter-stroke angles or stroke types (i.e., normal lines, zigzag lines, normal arcs, arcs formed by a set of circles, etc.). As we can see from these examples, the difference in stoke types may be subtle to distinguish. We do not distinguish concepts by the shape size and orientation, absolute position, or relative distance of two shapes.

## D.2 More Examples of Basic Shape Problems

(a) mountains

(b) nearly_full_moon and fish

(c) funnel and hourglass

(d) turtle and parallelogram

Figure 8: More examples from basic shape problems, where (a) the concept is mountains, (b) the concept is a combination of nearly_full_moon and fish, (c) the concept is a combination of funnel and hourglass, (d) the concept is a combination of turtle and parallelogram. In each problem, set $\mathcal{A}$ contains six images that satisfy the concept and set $\mathcal{B}$ contains six images that violate the concept. We also show two test images (left: positive, right: negative) in the binary classification problems. In basic shape problems, the task is about discovering the concept of the shape category itself. We do not distinguish concepts by the shape size and orientation, absolute position, or relative distance of two shapes.

## D.3 More Examples of Abstract Shape Problems

(a) `necked`

(b) `have_two_parts`

(c) `symmetric`

(d) `have_acute_angle`

Figure 9: More examples of abstract shape problems, where one attribute shared by the positive images in set $\mathcal{A}$ is considered as the underlying concept. In particular, (a) the concept is `necked`, (b) the concept is `have_two_parts` (it means the shape can be separated into two disconnected parts by a connecting point), (c) the concept is `symmetric`, (d) the concept is `have_acute_angle`. In each problem, set $\mathcal{A}$ contains six images that satisfy the concept and set $\mathcal{B}$ contains six images that violate the concept. We also show two test images (left: positive, right: negative) in the binary classification problems. We do not distinguish concepts by the shape size and orientation, absolute position, or relative distance of two shapes. We can see for abstract shape problems, it may also be challenging for humans without clearly understanding the meanings of abstract attributes.

(a) `self_transposed` and `exist_quadrangle`

(b) `balanced_two` and `exist_sector`

(c) `have_curve` and `closed_shape`

(d) `have_four_straight_lines` and
`thin_shape`

Figure 10: More examples of abstract shape problems, where a combination of two attributes shared by the positive images in set $\mathcal{A}$ is considered as the underlying concept. In particular, (a) the concept is a combination of `self_transposed` and `exist_quadrangle` ('self-transposed' means there is a central point, symmetrically around which every edge point of the shape could be mapped to another edge point), (b) the concept is a combination of `balanced_two` and `exist_sector` ('balanced' means the two parts in a shape have the similar area), (c) the concept is a combination of `have_curve` and `closed_shape`, (d) the concept is a combination of `have_four_straight_lines` and `thin_shape`. In each problem, set $\mathcal{A}$ contains six images that satisfy the concept and set $\mathcal{B}$ contains six images that violate the concept. We also show two test images (left: positive, right: negative) in the binary classification problems. We do not distinguish concepts by the shape size and orientation, absolute position, or relative distance of two shapes. We can see for abstract shape problems, it is also challenging for humans without clearly understanding the meanings of abstract attributes.