[Reviews · NeurIPS 2020]

Review 1

Summary and Contributions: This work introduces Bongard-LOGO, which is a new benchmark visual-reasoning-based dataset. Bongard problems have a number of advantages, namely: they involve context-dependent perception, demand strong analogical reasoning, and involve perception with an infinite vocabulary. The authors thoroughly develop a data generation process for Bongard problems that produces interesting, faithful examples that are challenging for contemporary visual processing models. They emphasize the role for quick, few-shot learning, as well as an ability to extrapolate to never-before-seen situations. Altogether, this is a very nice contribution to the space of machine-based visual reasoning, and the authors are commended for putting together a very lovely paper.

Strengths: I believe this is quite a strong paper that has many merits. The exposition is stellar, the methods are clear, the goals are stated succinctly, and they are met. The work is built on a solid footing from psychological sciences as well as contemporary machine learning. The empirical evaluation is adequate for a benchmark paper, and there is ample opportunity for future researchers to tackle and improve upon the results presented here. While I believe others have proposed Bongard style problems in recent times, I don't think any work has developed as rich of a dataset.

Weaknesses: There are very few weaknesses with this work. Perhaps the one omission I would have liked to see addressed is an assessment of whether the problems with current models have to do with visual acuity or reasoning per se. I ask because the shapes and lines in the panels are quite small, and fine, which may pose difficulties for visual models that have been optimized for images of a much different style. Do the authors have any baselines with models that process symbolic inputs of the problems?

Correctness: As far as I can tell, the claims and methods are technically correct.

Clarity: Yes, it is very well written. Great job!

Relation to Prior Work: Yes, past work is appropriately referenced.

Reproducibility: Yes

Additional Feedback: Congratulations for compiling this excellent paper. It was a pleasure to read. ##################################################### Update after reading rebuttals and other reviews: I'll maintain that this is a strong paper worthy of acceptance. I can agree with some of the other reviewers that some of the writing is a touch grandiose, and may have some unnecessarily strong claims, but I personally found it refreshing even if I did not agree with some of the points. The author's addressed my only qualm about the CNN-baseline, and I don't think there's any major deficiency that should cause the paper to be rejected.


Review 2

Summary and Contributions: The paper presents a new benchmark for visual reasoning and concept learning. It is directly inspired by the classical Bongard problems. The novelty is that is it procedurally generated, and of much larger size than the original Bongard problems. The paper includes an evaluation of existing methods, in particular from the few-short learning literature, since the task is formulated as a few-shot learning setup. UPDATE FOLLOWING REBUTTAL: The authors adequately answered my concern about the suitability of CNNs to line drawings, and I am happy to recommend the paper for acceptance. For their own benefit and that of the reader, I strongly suggest to briefly mention the few points of contention raised in the review (need for rotation invariance, relation with Raven-style tests including some those made of real images, verification of the suitability of CNNs to line drawings).

Strengths: The paper addresses a very interesting area that goes beyond the supervised/discriminative models for which deep learning has proven so successful. The benchmark adequately evaluates capabilities that SOTA models do not seem to possess. The evaluation includes a good number of existing methods.

Weaknesses: Although I appreciate the benchmark for the "concept learning" aspects that set it apart from most tasks in computer vision, I am skeptical about the type of stimuli used to measure these capabilities. The images are black-and-white drawings with fine lines. This makes the "visual recognition" part require different capabilities than what the state-of-the-art models for computer vision were designed for (natural images). The authors should first assess whether CNNs are suitable at all for this kind of visual input, regardless of the concept learning/reasoning aspects. This could be done by training a recognition task with a large training set of visual stimuli of this type (i.e. in a standard supervised setup). The authors state: "our benchmark aims to challenge the fundamental assumptions of objectivism and metaphysical realism held by most of standard image classification methods, demanding a new form of human-like perception that is context-dependent, analogical, and of infinite vocabulary" I don't see why realism and human-like perception are presented as being at odds. Maybe this was not the intention, and this could/should have been stated in two separate sentences ? Visual reasoning does not necessarily need synthetic data. It can be evaluated on realistic stimuli. See for example (not mentioned in the related work): V-PROM: A Benchmark for Visual Reasoning Using Visual Progressive Matrices, Teney et al., AAAI 2020 The task also requires perfect rotation invariance, which is not desirable in CNNs designed for object recognition in natural images. Specific methods have been developed though to provide this capability. They are not exploited here. This might bring a big improvement in the same was a CNNs provide built-in invariance to translation (and make such a difference vs a fully-connected network). Difference with RPMs I don't understand why/how this benchmark is needed (better ? complementary ?) compared to the few recent benchmarks using Raven-style tests (Barrett et al., Teney et al., Zhang et al.). L336 states: "our BONGARD-LOGO benchmark has posed a greater challenge", but the reported performance (50-60% range) is pretty high and in the same ballpark as the Raven-style benchmarks. Test sets requiring generalization The test sets are adequately designed to require some form of generalization, as explained in the text. But the level of generalization required is fixed with some fairly arbitrary choices. Other benchmarks like Barrett et al. and Teney et al. (mentioned above) were designed such that the level of generalization required could be varied, and models evaluated at multiple points. For example here, with free-form shapes, why settle on just +1 extra step than training programs ? This seems easy to vary. With abstract shapes, why choose "eight_straight_lines" as the held out attribute ? Others (even all others possibly) could have been tested in a held-one-out manner. Overall, the paper addresses an important, underserved area of research. The work seems to have been well conducted and the paper is well written. A few more experiments and discussion addressing the points raised above should make it suitable for publication at a top conference, and we encourage the authors to resubmit it at a future venue.

Correctness: Everything seems correct.

Clarity: The paper is reasonably well written, although it has a lot of repetitions. L36: "a myriad attempts": why only 2 citations ? That seems low for a myriad. L261: "3-5 human subjects": I don't understand the 3-5 range. Why is it not an exact number ?

Relation to Prior Work: See above.

Reproducibility: Yes

Additional Feedback:


Review 3

Summary and Contributions: In the 1960s, Bongard problems were introduced to test human-level concept perception with computational systems, but due to a number of reasons, this particular problem set has not seen widespread adoption in AI research. Two particular difficulties for training modern machine learning algorithms on these problems is that the original problem set is very small, consisting of 100 hand-curated problems, each with just 12 images, and that the solution consists of a verbal description of the discriminative concept. To help overcome these difficulties, while still maintaining the spirit of the original problems, the authors introduce a variant of this task with 12,000 individual problems that are procedurally generated, and where the solution is to be stated as a simple binary true/false classification.

Strengths: - Introduction of an easy-to-use Bongard-like task set with many sub-tasks to train and test on, which will be of particular interest for researchers interested in visual few-shot concept-acquisition learning problems - The tasks are grouped into four sub-categories, with baseline results for each of category - Importantly, one of the baselines is human performance, and nicely enough, humans seem to find the problems reasonably easy to solve - A good number of algorithmic baseline results are provided, giving a rough indication of the difficulty of the problems - I expect that this problem set will be picked up by a larger part of the research community that is interested in higher-level visual perception and visual concept learning

Weaknesses: - The way the problem set is generated and structured, the solutions favour some particular approaches over others, in particular program induction and meta-learning approaches: the entire setup is really a meta-learning setup, where the compositional structure is in the form of sequential drawing actions, as opposed to more general visual structure learning (the authors acknowledge a related point in the impact statement) - Related to the above, while Bongard problems are a specifc challenge problem to test some form of concept understanding that humans possess, it might be asking the wrong question to expect all the concepts to arise from the simple visuals in even this extended version. To make this point differently, while the core problem addressed here is providing a problem set for training these algorithms that acquire the concepts in question from the (Bongard-LOGO) data, this is unlikely the path how human perception proceeds, which is of course "trained" on much richer visual experience. - On a more specific note, the problem set seems actually limited even compared tot the original small problem set, e.g., no potential filling-in of enclosed areas, much more limited set of shapes, etc. - The manuscript makes strong claims in favor of particular viewpoints/approaches over others (eg., "deep learning" vs "neurosymbolic", as well as a number of strong philosophical claims that can be debated), even though this doesn't add to the contribution of the paper. I feel it should be left to researchers using the newly introduced tool to advocate their own interpretations of their preferred approaches, rather than potentially limiting the audience by appearing polarising. - More generally, I feel the paper espouses a rather biased angle on research in the field. As one example: "A myriad of attempts have been made in tackling the BPs with modern AI tools [12, 13]." -- I would consider it as vastly overstated that there have been myriads of attempts. - Finally, short of trying to understand all the implementational details and parameter searches for the baselines, I would take the them with a grain of salt as it is not necessarily in the interest of the authors to provide strong non-human baseline results. Having said that, I did not notice any particular issues, and have in fact been impressed by the due diligence that seems to have been applied in applying a number of different approaches.

Correctness: - The task setup, while limited (see comments above) seems sound, and transforming the language-based Bongard problem solutions into a binary classification problem is a creative solution

Clarity: - The paper is well-structured, well written, and easy to follow, notwithstanding that I found myself disagreeing with a number of general claims that seem peripheral to the core contributions of the paper.

Relation to Prior Work: - Clearly explained and motivated why the authors saw the need to introduce this new task set, and how it can facilitate and open up more research in this particular set of questions that Bongard problems aim to address.

Reproducibility: Yes

Additional Feedback:


Review 4

Summary and Contributions: This paper introduces a new challenge problem for combinatorial concept learning. It is a modified version of the original Bongard problem so that it is compatible with data-driven methods. The core hypothesis is to test the ability of representation learning methods to solve a set of hold out bongard problems where the challenges arise due to the need to perform joint shape, analogical, compositional and context reasoning. This seems like a good challenge problem for deep learning based meta learning methods as the gap between humans and algorithms is still quite large. ------------------------ After rebuttal ----------------- The authors addressed most of the concerns raised during the review period. For this reason, I am raising my score for this paper.

Strengths: The dataset introduced in this paper is interesting, hard and can generate interesting follow up research. It will facilitate research into investigating the type of neural network architectures and loss functions that best serve use cases which require joint analogical and combinatorial reasoning. This is still a big open problem in representation learning and differentiable reasoning. Also, compared to other differentiable symbolic reasoning tasks, this task has an added dimension of visual reasoning. The empirical experiments are reasonable and the human baseline is also established. This gives a good sense for the difficulty of the task and for other follow up work to compare against. This paper also shows that meta learning methods perform better on the tasks than their counter parts. So this might be a productive and simple playground for meta learning approaches to improve.

Weaknesses: "our benchmark aims to challenge the fundamental assumptions of objectivism and metaphysical realism held by most of standard image classification methods, demanding a new form of human-like perception that is context-dependent, analogical, and of infinite vocabulary" - The paper is written in an overly grandiose tone. I am not sure if the experiments clearly demonstrates any meta physical point -- it is simply pointing out the challenges in combinatorial reasoning in data-driven representation learning. The paper shows that meta learning helps in solving the task but does not say much about model parametrization. The core concept learning problem is about contextual compression and it would have been very interesting to run ablation experiments on model sizes and observe empirical performance. How does under or over parametrization or other information bottlenecks in the architecture lead to generalization? "A potential risk of our new benchmark is that it might skew research towards certain directions, e.g,. inductive logic programming methods designed for solving Bongard Problems" - If ILP is the solution to these problems then this statement sounds paradoxical as why shouldn't one combine ILP with data-driven methods? This also another criticism of this paper. If you applied hand-crafted methods then what is the performance on the benchmarks? It is interesting to see modern methods perform this task but it is also important and interesting to know how existing methods perform (irrespective of how they generalize to other problems).

Correctness: The dataset and experiments are interesting and reasonable. I would have liked to see how older methods (e.g. ILP) perform on their datasets.

Clarity: This paper is generally clear and well written. However, it also has a grandiose tone to it -- especially in the introduction.

Relation to Prior Work: Prior work to tackle the original bongard problems (ILP and older methods) are discussed but none of these works were included in the empirical evaluation.

Reproducibility: Yes

Additional Feedback:

[Author Response · NeurIPS 2020]

We thank all the reviewers for helpful comments and we address the major concerns in the following.

**(R1, R2) The use of CNN backbones in our visual reasoning tasks.** The CNN backbone is capable of processing
the visual stimuli of our BONGARD-LOGO tasks. We first demonstrated this in the ablation study in Appendix, Section
C, where Table 2 shows that the best train/test accuracies on free-form shapes are 97.6% and 89.0%, respectively. This
implies that isolating from the concept reasoning properties of *context-dependent* and *analogy-making* perception, the
CNN backbones can achieve high recognition performance on our visual inputs. Furthermore, we performed standard
supervised experiments on human designed shapes by training two separate binary attribute classifiers for `convex` and
`have_two_parts`, respectively, using the same CNN backbone as in the paper. With each dataset of randomly sampled
14K positive and 14K negative samples, the model achieved the near-perfect train/test accuracies 98.7%/97.0% on
`convex` and 98.0%/97.1% on `have_two_parts`.

**(R2) Differences with RPMs.** Our benchmark is complementary to RPMs: RPMs focus on relational concepts (such as
progression, XOR, etc.) while our BONGARD-LOGO problems focus on abstract object concepts (such as stroke types,
abstract attributes, etc.). In RPMs, the relational concepts come from a small set of five relations [17]. In our benchmark,
the object concepts can vary arbitrarily with procedural generation. The three core properties (i.e., context-based,
analogy-making, few-shot with infinite vocabulary) of human cognition captured by our benchmark also define a new
challenge for current object perception methods. Besides, the model performance of 50-60% in our benchmark is not as
high as it might seem. Each of our tasks asks for a binary decision, where chance performance is 50%. The majority of
the model performances in Table 1 are within 10% better than chance. In contrast, the model performance of 60-70%
accuracy achieved on RPMs [17] is much higher than its chance performance of 12.5% accuracy.

**(R2) Visual reasoning on V-PROM and real images.** Thank you for bringing up V-PROM, an interesting work on
extending RPMs to real images. As V-PROM is a real image version of RPMs, it differs from our benchmark in the
similar ways described above. We chose synthetic data because a) Bongard designed the original problem sets with
simple line drawings and geometries, and we follow the same design principle. b) Furthermore, procedural generation of
synthetic data gives us the precise control of concepts and nuisances, making a systematic study of model performance
on Bongard problems feasible. Some early attempts of building our benchmark were indeed based on real-world images,
such as CelebA and MS-COCO datasets. However, the quality is hindered by uncontrollable confounders in real images,
resulting in ambiguous concepts.

**(R2) The level of generalization in the test sets.** Similar to prior work, the level of generalization in our benchmark
can be varied flexibly, and models can be evaluated at multiple points. We chose "+1 extra action stroke" and "+1 more
straight line" in the held-out sets as two basic cases for extrapolation. As these two tasks have been shown difficult for
current methods (i.e. performances close to chance in Test Acc (FF) and (NV) as shown in Table 1), we did not include
more challenging setups. We will expand our discussions on this in the future revision.

**(R2, R3, R4) Strong philosophical claims.** The statements of objectivism in traditional AI [14] and metaphysical
realism in Bongard problems [15] are referenced from prior work. We included them to highlight the intellectual roots
of our work and its connections to other disciplines. We concur with the reviewers and we will tone down our language,
focusing on the technical merits of this work in the future revision.

**(R3) Problem set in BONGARD-LOGO versus the original Bongard problem set.** Taking advantage of procedural
generation, we are able to produce an infinite amount of free-form shapes, which the original set does not have. The
procedural generation method can be easily extended to incorporate new shape features, including filling-in of enclosed
areas. We will also open-source our procedural generation code for the research community to create their own shapes.

**(R4) Ablation study on model sizes versus performances.** We vary the model size by dividing each layer size in the
ResNet-15 backbone by a reduction factor $\alpha$ ($\alpha$ is a divisor of the number of parameters). Figure 1 illustrates the results
of two top-performing models: ProtoNet and Meta-Baseline-SC. We see that Train Acc decreases consistently with
smaller model sizes. On the test sets, results slightly vary across different models. ProtoNet is more sensitive to model
size than Meta-Baseline-SC: All the Test Acc of ProtoNet tend to decrease with smaller model sizes, while some Test
Acc (BA and CM) of Meta-Baseline-SC remain robust. We will add these ablation results in the future revision.

Figure 1: Performance of Meta-Baseline-SC and ProtoNet under different model sizes (model size is smaller with larger $\alpha$).

**(R4) Traditional ILP methods for BONGARD-LOGO tasks.** ILP does not offer complete solutions to Bongard
problems. The best ILP method so far [13] has relied on manually specified rules to tackle a subset of 39 carefully
selected Bongard problems, making it infeasible for our benchmark which consists of tens of thousands of problems.
We intentionally avoid highly customized methods that may have low generality and limited applicability, which deviate
from the motivation of our benchmark as a step towards inspiring more robust and reliable AI systems. That being
said, we very much agree with R4 that a *hybrid* system that combines data-driven and symbolic methods is a promising
future direction for both generality and reliability, as we discussed in Section 5.

[Meta-Review · NeurIPS 2020]

Overall reviews agree that the paper and the dataset proposed with this paper is interesting and valuable. Though some of the reviewers think that the writing is somewhat grandiose, and some of the claims are unnecessarily strong. It would be nice if the authors can address those issues for the camera-ready.